# Bridging the Gender Gap in the Agricultural Sector: Evidence from European Union Countries

Rosa Maria Fanelli 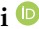

Department of Economics, University of Molise, 86100 Campobasso, Italy; rfanelli@unimol.it

**Abstract:** In European Union countries most farms are operated by smallholders in extensive agriculture and approximately 30% of agricultural farms are operated by woman. Despite the growing importance of the role of women in the agricultural sector, a more complete picture of the gender gap and the differences between female and male-operated farms is still lacking. The principal aim of this paper is to fill this gap by highlighting the differences between female and male-operated farms. The data used comes from the World Bank Open Data repository and Eurostat. The methodology consists of descriptive statistical analysis, principal component analysis, and multivariate regression models. The analysis is used to investigate how a set of indicators affects the gender gap in the agricultural sector. The research questions test how the socio-economic determinants and traditional agricultural land use influence the historical gender division of labour and the evolution and persistence of farm differences. The results from almost all 27 EU countries suggest that women, the same as men, can be considered "productive resources" and play an important role in the agriculture sector in areas such as crop and livestock production. However, their activities are less remunerative.

**Keywords:** agricultural sector; gender gap; multivariate regression models; principal component analysis





## 1. Introduction

The principal aim of this paper is to present the results of a study into the gender gap (gender inequalities) that influences the characteristics of the agricultural sector. It focusses specifically on how the sector is affected by the different determinants of the economic systems of European Union (EU) countries and by diverse agricultural land use, which very little is known about. In recent years, there has been a growing political–institutional interest in the issues related to the presence and contribution of women in agriculture. The most recent FAO statistics highlight that woman comprise about 43% of the agricultural labour force globally. In developing countries, this figure ranges from 20% in Latin America to 50% in Eastern Asia and Sub-Saharan Africa. Indeed, there is considerable variation across EU countries according to their socio-economic and labour market determinants, and their diverse agricultural land use.

The contribution of women to agricultural and food production is significant, but a thorough investigation of the differences between the realities for women and men in this sector is needed (FAO 2011). The activities of women represent a crucial resource in agriculture and in the rural economy. Women adopt roles as farmers, labourers, and entrepreneurs, and their activities typically include producing agricultural crops, tending to animals, processing and preparing food, working for wages in agricultural or other rural enterprises, engaging in trade and marketing, caring for family members, and maintaining their homes (FAO 2011).

At the EU level (EU-28), around 30% of agricultural farms are operated by women—one farm in five, and in rural areas of the EU women represent over 50% of the rural population.

Women comprise about 45% of the total working population and about 35% of workers in the agricultural sector of the EU-28, the work of female farmers accounts for 31% of the total working hours (European Parliament 2019). Despite this importance, little is known about the differences between the characteristics of farms managed by women and those managed by men.

In the previous literature, gender inequalities in farming activities have been analysed from different perspectives, using different theoretical approaches with the aim of highlighting the role of women in agriculture and in rural contexts (Sachs 1983; Errington and Gasson 1993; Little 2006; Fanelli 2020).

In order to fill the research gap identified above and to widen the debate surrounding this subject, the principal aim of this study is to explore the gender differences in the agricultural sector of the 27 EU countries from the perspective of the role of women in agriculture (Whatmore 1994).

To validate this framework, the main research questions (RQs) are as follows:

$RQ_1$: Are the interaction effects of the socio-economic and labour market determinants on the gender gap in the agricultural sector significant?
$RQ_2$: Are there significant differences in the use of agricultural land between EU countries?
$RQ_3$: To what extent are the characteristics of farms run by women affected by different socio-economic and labour market determinants?
$RQ_4$: How can the gender gap in the agricultural sectors of EU countries be bridged?

The paper is structured as follows: Section 2 puts forward a brief theoretical framework on gender studies in agriculture; the aim is not to provide a complete literature review but to focus on the key determinants the empirical analysis is centred around. Section 3 reports on the construction of a conceptual model of the gender gap and describes the data collection and methods, while Section 4 is devoted to the main results obtained. Finally, Section 5 concentrates on a discussion of the results, the conclusions drawn, and suggestions for future research.

## 2. Gender Studies in Agriculture: A Theoretical Framework

In Europe, scientific research on gender studies in agriculture has developed in different strands that analyse the process of affirmation of women in agriculture, the structural and dimensional characteristics of farms run by women, the role of women in multifunctional enterprises, and the relationship between women and social capital in agriculture. The process of affirming the role of women in agriculture was driven initially by the economic development of the 1970s, which, as is well known, led to a rural exodus and the abandonment of agriculture by the young, male workforce. Women replaced men in the context of economic marginalisation of corporate resources, especially in the rural areas of the EU. The presence of women in agriculture is analysed both in a "quantitative" way, highlighting the growing relative presence of women in the agricultural sector, and from a different perspective, which examines the opportunities that the new agri-food scenarios offer the female workforce. There seems to be sufficient uniformity of views on the fact that the evolution of the rural world, the growing importance of multifunctional agriculture, and economic diversification, resulting from the rural development model advocated by EU policies, constitute opportunities for the affirmation of the role of women in agriculture (Gidarakou 1999).

Some studies have investigated the organisational, social, and economic aspects of agriculture for women (Boeraeve-Derijke 1994). The contribution of women to agriculture has in many cases been described as substantial but "invisible", which is due mainly to lack of any registration in the statistics. Consequently, an entire line of studies has been undertaken precisely with the aim of "making women visible in terms of hours worked, tasks assigned and involvement in the decision-making phases of the activity" (Whatmore 1994; Fieldsend 2008).

An attempt has been made to highlight the problems that may arise not only due to the different roles that women can cover in the family business, as conductors, assistants,

simple employees or other roles within the family (Vadnaja and Zupan 2009), but also in relation to the economic dimensions of the farm, the production organisation, the technologies adopted, and the territory in which the farms operate (Montresor 1994). The female sex has often been regarded as a natural representative of the rural world (Barlett et al. 1999). Furthermore, in recent years, several empirical studies have been conducted to analyse the magnitude of the earnings gap between women and men. The findings of these studies suggest a declining trend in gender wage discrimination since the 1960s in industrial countries, as a result of factors such as progress in women's education, increased job experiences, and the introduction of non-discrimination regulations (Weichselbaumer and Winter-Ebmer 2007). In contrast, there has been a growing debate regarding the extent to which women workers have benefited from recent trends in the global economy.

Other studies have highlighted the existence of a relationship between education and agricultural productivity (Appleton and Balihuta 1996; Asadullah and Rahman 2009; Lockheed et al. 1980; Pudasaini 1983; Weir 1999). Research shows that the returns of education on agricultural productivity vary depending on the educational level obtained (primary, secondary, and tertiary levels of education). Furthermore, higher education leads to greater participation for women in the labour market (Ahituv and Lerman 2007; Benavot 1989; Carnoy 2006; Crompton et al. 2007; Mok 2016; Woodd 2013).

Regarding the presence of women in the agricultural sector, the share of women workers is only 9.5% in upper-middle-income countries and 2.6% in high-income countries, while agriculture remains the most important employment sector for women in low-income and lower-middle-income countries (United Nations 2018, E/CN.6/2018/3). Land is perhaps the most important economic asset; women account for only 12.8% of agricultural landholders in the world (FAO 2015). In all EU member states, employment rates for women are lower than those for men, with big variations across the EU. Globally, women remain less likely to participate in the labour market than men; they are more likely to be unemployed than men; and they are over-represented in informal and vulnerable employment. Female employment rates depend on numerous socio-demographic and cultural factors, as well as the economic and political environment in individual member states (World Bank 2012; World Bank and IFPRI 2010). Women also bear disproportionate responsibility for unpaid care and domestic work. In recent years, in some EU regions a new phenomenon has come about, which shows that well-educated women are deciding to move to the countryside to carry out their professional activities (European Parliament 2019).

## 3. Materials and Methods

### 3.1. A Conceptual Model of the Gender Gap

To develop conceptual country-level models of the differences between the involvement of men and women in farming, three major sets of determinants were draw from the existing literature:

- The key socio-economic and labour market determinants that provide evidence of the differences between men and women in each country of the EU;
- The agricultural land use that highlights the availability of agricultural land, the main production systems, and the added value obtained from the agricultural sector (including forestry and fishing);
- The characteristics of the agricultural sector.

#### 3.1.1. The Key Socio-Economic and Labour Market Determinants

According to the existing literature, the gender gap persists in economic systems in regard to nine key differences: the synthetic index on gender equality (GEI), that tests how EU member states have different goals in terms of achieving equality between men and women; the different levels of education of women and men (F-EA.TE vs. M-EA.TE); the female and male participation rate (FLPR vs. MLPR); how women and men contribute to work in the family (SL.FAM.WORK.FE.ZS vs. SL.FAM.WORK.MA.ZS); the rates of female and male employment in the industrial (SL.IND.EMPL.FE.ZS vs. SL.IND.EMPL.MA.ZS) and service

sectors of the local community (SL.SRV.EMPL.FE.ZS vs. SL.SRV.EMPL.MA.ZS); the rates of female and male unemployment (SL.EMP.WORK.FE.ZS vs. SL.EMP.WORK.MA.ZS); the rates of male and female employment in waged and salaried jobs (SL.UEM.TOTL.FE.ZS vs. SL.UEM.TOTL.MA.ZS); the differences between women and men in terms of life expectancy at birth (SP.DYN.LE00.FE.IN vs. SP.DYN.LE00.MA.IN); and the percentage of the rural population in a country (SP.RUR.TOTL.ZS).

Regarding the GEI, a negative sign on the regression model is expected; if there were gender equality in a country this would encourage women to participate in higher wage-earning sectors such as the industrial and services sectors rather than in the agricultural sector. Special attention is paid to the interactions between men and women in relation to male and female participation and unemployment. It might be that men and women compete for the same jobs, but it could also be that the labour market status of men (especially with regard to male unemployment) has a discouraging effect on the behaviour of women or vice versa.

### 3.1.2. Agricultural Land Use

The second set includes variables representing the features of the agricultural land in which farms run by women and those run by men are operating. Land is the most important agricultural production factor; a high share of agricultural (AG.LND.AGRI.ZS) and arable land (AG.LND.ARBL.ZS) on the total land in a county provides both male and female farmers with more opportunities to set up different agricultural activities.

Furthermore, some types of agriculture such as land under cereal production (AG.-LND.CREL.ZS), permanent crops (AG.LND.CROP.ZS), and forest area (AG.LND.FRST.K3) may be carried out more effectively by women, or vice versa.

Finally, the share of value added (NV.AGR.TOTL.ZS) by agriculture (including forestry and fishing) on the gross domestic product (GDP) in a country indicates not only the importance of this sector for the economic system of the same country but also its level of development.

### 3.1.3. The Characteristics of the Agricultural Sector: Female vs. Male

The third category of variables describes the characteristics of the agricultural sector in the 27 EU countries such as: the share of farms operated by women and that of farms operated by men (F-OF vs. M-OF); the share of the agricultural area that women and men utilise for their activities (F-UAA vs. M-UAA); the share of female and male employment in the agricultural sector (SL.AGR.EMPL.FE.ZS vs. SL.AGR.EMPL.MA.ZS); the share of the farm area with the exclusion of special agricultural production areas (F-FAESAPA vs. M-FAESAPA); the share of farms with livestock (F-FWL vs. M-FWL); the units of live livestock (F-FWLL vs. M-FWLL); the share of agricultural standard output on the total (F-SO vs. M-SO); and the share of farms whose households consume more than 50% of the final production (F-FWHC > 50% vs. M-FWHC > 50%).

### 3.2. Study Area and Data

The data used for this study are drawn from the World Bank Open Data repository (World Bank 2019), Eurostat (Eurostat 2016) and the European Institute for Gender Equality (Eige 2019). The study is based on data for the 27 member states of the EU: Austria, Bulgaria, Cyprus, the Czech Republic, Germany, Denmark, Estonia, Greece, Spain, Finland, France, Hungary, Italy, Lithuania, Luxembourg, Latvia, the Netherlands, Poland, Portugal, Romania, Sweden, Slovenia, Slovakia, and the United Kingdom. The variables used, which relate to the selected socio-economic and labour market determinants on agricultural land use and on the characteristics of the agricultural sector, are described in the above sub-section. In the analysis, the shares rather than the absolute numbers of variables are considered. This is because attention is given to those factors associated with the major or minor presence of women and men in each agricultural sector of the 27 EU countries. All analyses were carried out by means of the STATA program (version 12.32).

### 3.3. Principal Component Analysis

The general purpose of principal component analysis (PCA) is to find a way of condensing the information contained in a huge number of original variables into a smaller set of new composite factors with minimum loss of information (Çamdevýren et al. 2005; Pires et al. 2008). By using this multivariate statistical method, input variables change into principal components (PCs) that are an independent and linear compound of input variables (Lu et al. 2003). In this analysis, PCA was carried out to obtain the latent root (Eigenvalue) from which the PCs were extracted. This was conducted to understand the differences between the involvement of women and men in the agricultural sector. Prior communality estimates with eigenvalues higher than 1.0 (Kaiser 1958) are set to one and components with an eigenvalue smaller than one are dropped from the analysis (Braeken and van Assen 2017). A varimax variable rotation is used to arrive at the PCs presented in the results section. Significant loadings on PCs are defined as those with a loading greater than 0.30 in absolute value. The final aim with the application of the PCA is to reduce the number of possibly correlated variables into a smaller number of uncorrelated new variables called PCs, which are orthogonal to each other. The PCs are designated as factors, which are estimated as a weighted sum of the original input variables.

### 3.4. Multivariate Regression Models

Multivariate regression analysis (MRA) was employed to better understand the associations between the key determinants of the principal difference between women and men in terms of their use of the agricultural land and the characteristics of the agricultural sector. This is one of the most commonly used techniques in the related gender gap literature (Peterman et al. 2014). The new variables obtained from the PCA are used as independent variables to set up two multivariate regression models (MRMs), since they optimise spatial patterns and remove possible complications caused by multicollinearity phenomena (Katrutsa and Strijov 2017). They also identify the factors that are the most significant in making the prediction (Çamdevýren et al. 2005; Sousa et al. 2007). In the two MRMs, the eight variables that describe the characteristics of the agricultural sector represent the dependent variables and serve to analyse the different situation in a country (*i*) in 2016 for women (1) and for men (2) in the same sector.

$$F\text{-}OFi_{2016}, F\text{-}UAAi_{2016}, SL.AGR.EMPL.FE.ZSi_{2016}, F\text{-}FAESAPAi_{2016}, F\text{-}FWLi_{2016},$$
$$F\text{-}FWLLi_{2016}, F\text{-}SOi_{2016}, F\text{-}FWHC > 50\%i_{2016} = PC_1 + PC_2 + PC_3 \ldots \ldots \ldots PCn \tag{1}$$

where vectors $PC_1$, $PC_2$, and $PC_3 \ldots \ldots \ldots PCn$ refer to the new latent dimension for the socio-economic and labour market determinants and for agricultural land use obtained with the application of PCA. This step is taken to summarise the information relating to the situation for women.

$$M\text{-}OFi_{2016}, M\text{-}UAAi_{2016}, SL.AGR.EMPL.MA.Zsi_{2016}, M\text{-}FAESAPAi_{2016}, M\text{-}FWLi_{2016},$$
$$M\text{-}FWLLi_{2016}, M\text{-}Soi_{2016}, M\text{-}FWHC > 50\%i_{2016} = PC_1 + PC_2 + PC_3 \ldots \ldots \ldots PCn \tag{2}$$

where vectors $PC_1$, $PC_2$, and $PC_3 \ldots \ldots \ldots PCn$ refer to the new latent dimension for the socio-economic and labour market determinants and for agricultural land use obtained with the application of PCA, this time to summarise the information relating to the situation for men.

## 4. Results

### 4.1. Descriptive Statistics

This sub-section reports the general descriptive results for female and male farmers, while the next sub-session reports the PCA and MRM results. Both the share of female- and male-operated farms from 2005 to 2016 fell in the EU (Table 1).

**Table 1.** Female- vs. male-operated farms in the EU countries, var % 2016–2005.

| Countries | Female Operated Farms | Male Operated Farms | Total Farms |
|---|---|---|---|
| Belgium | −31.01 | −27.97 | −28.42 |
| Bulgaria | −46.83 | −65.42 | −62.13 |
| Czechia | −54.74 | −33.71 | −37.23 |
| Denmark | −53.69 | −29.45 | −32.19 |
| Germany | −20.87 | −29.96 | −29.18 |
| Estonia | −46.10 | −36.13 | −39.82 |
| Ireland | −1.65 | 4.37 | 3.68 |
| Greece | −10.47 | −20.32 | −17.84 |
| Spain | 3.86 | −16.66 | −12.76 |
| France | −20.53 | −19.48 | −19.70 |
| Italy | −25.16 | −37.04 | −33.72 |
| Cyprus | −18.15 | −23.84 | −22.63 |
| Latvia | −47.92 | −43.63 | −45.64 |
| Lithuania | −37.22 | −43.06 | −40.58 |
| Luxembourg | 9.68 | −23.83 | −19.59 |
| Hungary | −30.66 | −42.69 | −39.84 |
| Malta | −38.46 | −13.86 | −15.88 |
| Netherlands | −47.36 | −30.85 | −31.96 |
| Austria | −29.36 | −20.90 | −23.79 |
| Poland | −47.54 | −40.92 | −43.04 |
| Portugal | −2.41 | −26.05 | −20.24 |
| Romania | −6.85 | −24.88 | −19.66 |
| Slovenia | −30.00 | −2.12 | −9.42 |
| Slovakia | −63.31 | −62.37 | −62.55 |
| Finland | −19.95 | −30.71 | −29.59 |
| Sweden | 12.60 | −20.79 | −16.98 |
| United Kingdom | −45.83 | −33.24 | −35.51 |
| Total | −22.84 | −30.82 | −28.72 |

Source: own calculations based on Eurostat and Word Bank data.

Farms operated by women declined by more than 50% in Slovakia (−63.31%), Czechia (−54.74%), and Denmark (−53.69%), while farms run by men declined in Bulgaria (−65.42%) and Slovakia (−62.37%).

Only three countries recorded positive trends for female-run farms compared with those run by men. The former increased by 13% in Sweden, 10% in Luxembourg, and 4% in Spain, while the latter decreased by 21%, 24%, and 17%, respectively, in the same countries. Only Ireland recorded an increase of more than 4% in male-run farms. Overall, from 2005 to 2016, farms operating within the territory of the EU decreased by almost 29%.

Male-run farms (−31%) rather than female-run farms (−23%) contributed more to this decline. It is possible to hypothesize that in the period of 2005–2016 farms run by women showed greater resilience compared with those run by men.

As can be seen in Figure 1, the EU countries with a higher incidence of female-run farms (>30%) on the total number of farms are in order of importance: Latvia, Lithuania, Romania, Estonia, Austria, Italy, and Portugal. Conversely, those with a low incidence (<10%) are the Netherlands, Malta, Denmark, and Germany. This can be attributed to the fact that in the first block of countries, especially in Latvia and Lithuania, subsistence agriculture prevails, and farms encounter the marginalisation of productive resources and human capital; while in the second block, agriculture with market-oriented farms prevails, has higher incomes, and is led mainly by men. Furthermore, these trends are related not only to cultural factors but also, and above all, to the type of agricultural activity present in the various geographical areas. So much so that the states with the lowest quotas of female management (<10%) are all characterised by a strong zootechnical specialisation. This has traditionally been the prerogative of the male gender, since the establishment of the first

complex and technologically advanced communities ("intensive agriculture passes into the hands of men and breeding is always a male task" (Arioti 1983)).

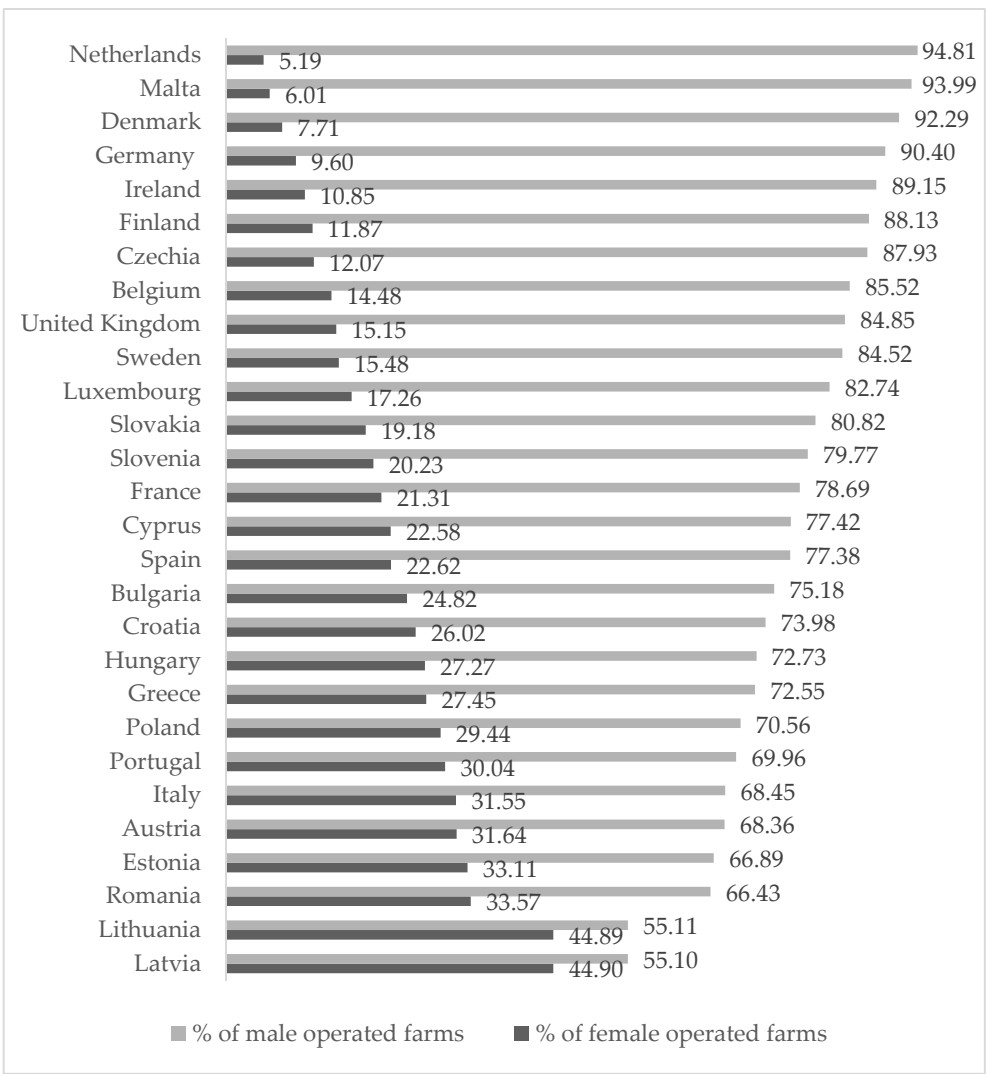

**Figure 1.** Female- vs. male-operated farms (% on the total). Source: own calculations based on Eurostat and Word Bank data.

RQ$_1$: Are the interaction effects of the socio-economic and labour market determinants on the gender gap in the agricultural sector significant?

The results obtained with the application of descriptive statistical analysis (DSA) highlight that the GEI, on average, is about 65%. The average scores of the GEI range from 51.2% in Greece to 83.6% in Sweden (Eige 2019). Regarding the educational attainment in the analysis, only those with tertiary education were considered. This indicator highlights that, on average, the percentage of women with this level of education is greater (44% vs. 25%). However, the labour participation rate is higher for males than for females (54% vs. 46%), especially in the service sector that is more important for women than for men in terms of employment (83.7% vs. 60.5%). There is always a significant gap between the level of male and female employment, and for women the service sector is relatively more important than the industrial sector in terms of employment opportunities. The EU member states present, on the basis of the value of coefficient of variation (CV), a homogenous situation regarding the labour participation rate for both sexes. For women, the situation in EU countries ranges from a minimum of 40.56% in Malta to a maximum of 50.53% in Lithuania. Working women, however, contribute more to housework than men (1.32% vs.

0.67%). However, the high value of the CV shows that this contribution varies substantially across EU countries. In addition, on average, the unemployment rate is higher for women than for men (89% vs. 82%). Data show that working women were generally more often salaried than their male counterparts (6.7% vs. 6.3%), but important differences, as shown by the values of the CV, exist in the number of female salaried workers from country to country with respect to the number of men. These differences reflect the distribution of jobs between different sectors of the economy, since women tend to be concentrated in the tertiary sector. On the other hand, the life expectancy at birth, on average, is higher for women than for men (83 vs. 77 years). Finally, on average, just over one quarter (26%) of the EU population lived in a rural area in 2016, with some large dissimilarities between the 27 EU countries. The percentage varies from a minimum of 2% in Belgium to a maximum of 46% in Slovakia (Table 2).

**Table 2.** Summery statistics of the items used in evaluating the differences among female and male.

| | Female | | | | |
|---|---|---|---|---|---|
| **Variables** | **Mean** | **Std. Dev.** | **Min** | **Max** | **Coef. of var.** |
| GEI | 64.59 | 8.7 | 51.2 | 83.6 | 13.47 |
| F-EA.TE | 43.56 | 9.76 | 25.7 | 63.4 | 22.41 |
| FLPR | 46.38 | 2.42 | 40.56 | 50.53 | 5.22 |
| SL.FAM.WORK.FE.ZS | 1.32 | 2.30 | 0.10 | 11.65 | 174.24 |
| SL.IND.EMPL.FE.ZS | 12.92 | 5.81 | 4.11 | 23.93 | 44.97 |
| SL.SRV.EMPL.FE.ZS | 83.70 | 8.63 | 56.34 | 95.43 | 10.31 |
| SL.EMP.WORK.FE.ZS | 6.74 | 4.61 | 2.8 | 24.29 | 68.4 |
| SL.UEM.TOTL.FE.ZS | 89.00 | 4.78 | 73.46 | 94.62 | 5.37 |
| SP.DYN.LE00.FE.IN | 83.03 | 2.09 | 78.6 | 86.3 | 2.52 |
| SP.RUR.TOTL.ZS | 25.65 | 12.86 | 2.00 | 46.27 | 50.14 |
| | Male | | | | |
| **Variables** | **Mean** | **Std. Dev.** | **Min** | **Max** | **Coef. of var.** |
| GEI | 64.59 | 8.7 | 51.2 | 83.6 | 13.47 |
| M-EA.TE | 24.78 | 10.45 | 11.2 | 54.7 | 42.17 |
| MLPR | 53.62 | 2.41 | 49.55 | 59.44 | 4.49 |
| SL.FAM.WORK.MA.ZS | 0.67 | 0.86 | 0.07 | 4.28 | 128.36 |
| SL.IND.EMPL.MA.ZS | 33.71 | 8.05 | 18.21 | 49.6 | 23.88 |
| SL.SRV.EMPL.MA.ZS | 60.50 | 9.81 | 41.01 | 80.24 | 16.21 |
| SL.EMP.WORK.MA.ZS | 6.30 | 3.02 | 1.79 | 15.41 | 47.94 |
| SL.UEM.TOTL.MA.ZS | 82.14 | 5.54 | 64.29 | 90.38 | 6.74 |
| SP.DYN.LE00.MA.IN | 77.35 | 3.55 | 70.10 | 81.2 | 4.59 |
| SP.RUR.TOTL.ZS | 25.65 | 12.86 | 2.00 | 46.27 | 50.14 |

Source: own calculations based on Eurostat and Word Bank data.

RQ$_2$: Are there significant differences in the use of agricultural land between the EU countries?

On average, in the 27 countries of the EU, the agricultural area accounts for almost 43% of the total area. The country with the lowest incidence is Sweden (7%), followed by Finland (7.5%), and Cyprus (14%), while the country with the highest incidence is the United Kingdom (72%), followed by Denmark (66%), and Ireland (66%). On the other hand, in relation to arable area, the value of the CV (51%) highlights a greater heterogeneity between EU countries with respect to the incidence of the agricultural area on the total (39%). On average, within the EU, 25% of the area is arable, and the situation varies from a minimum (6.3%) in Sweden to a maximum (60%) in Denmark. However, in relation to the cultivation systems, the greatest dissimilarity (with a CV value equal to 135%) among the 27 EU countries is found in the incidence of the area invested in permanent crops (vineyards, olive groves and orchards) that, on average, represents almost 3% of the total area of the EU-27, with peaks of 0.01% for the Nordic countries (Finland, Ireland, and Sweden) and with peaks ranging from 9.8% to 8% in Mediterranean countries (Spain,

Portugal, Greece, and Italy). On the one hand, the Nordic states (Finland and Sweden but also Slovenia) have a greater incidence of forestry area on the total area. On the other hand, Denmark and Eastern European countries (Hungary, Poland, Romania, and Lithuania) have a greater incidence when compared with the area devoted to cereals. The forestry area, that, on average, represents 34% of the surface of the EU-27, shows less heterogeneity between countries than the average of 12% invested in cereals. Finally, the added value of the agricultural, forestry, and fish sectors accounts for an average of almost 2% of the GDP. The minimum value (0.23%) is recorded for Luxembourg while the maximum (4.3%) is for Romania. The value of the CV attests that there is heterogeneity between the 27 EU countries, and in particular between the poorest countries of the EU (Romania, Greece, Latvia, Hungary, and Bulgaria) and the richest nations (Luxembourg, Belgium, the United Kingdom, and Germany). The agricultural sector, therefore, continues to represent the most important economic sector for low-income countries (Table 3).

**Table 3.** Summery statistics of the items used in evaluating the agricultural land-use determinants.

| Variables | Mean | Std. Dev. | Min | Max | Coef. of var. |
|---|---|---|---|---|---|
| AG.LND.AGRI.ZS | 42.85 | 16.58 | 7.39 | 71.72 | 38.69 |
| AG.LND.ARBL.ZS | 25.08 | 12.86 | 6.26 | 59.8 | 51.28 |
| AG.LND.CREL.ZS | 12.48 | 8.38 | 2.06 | 35.41 | 67.15 |
| AG.LND.CROP.ZS | 2.19 | 2.95 | 0.01 | 9.78 | 134.7 |
| AG.LND.FRST.K3 | 34.10 | 17.69 | 1.44 | 73.73 | 51.88 |
| NV.AGR.TOTL.ZS | 1.98 | 1.09 | 0.23 | 4.33 | 55.05 |

Source: own calculations based on Eurostat and Word Bank data.

The third set of variables describes the characteristics of the agricultural sector in the 27 EU countries. The data reported in Table 4 show that, on average, more than three quarters of farms are run by men (78%). These farms are generally larger than those run by women as they incorporate, on average, more than 86% of the agricultural area used for production purposes (UAA). Agriculture tends to be much more important for men than for women in terms of the percentage of employment (5.79% vs. 3.39%). This importance, as shown by the high values of the CV, varies substantially across countries, especially for women. About 80% of farms with livestock are run by male farm holders, while only 21% are run by female farm holders. The differences are greater with regard to the units of live livestock (87% for male vs. 16% for female). Differences between men and women in the average area of farmland owned are also reflected in the output per holding, where women farmers also fare much less well than their male counterparts (0.15% vs. 84%). Finally, there is an evident gender gap in the farms whose household consumption exceeds 50% of final production (40% vs. 19%) (Table 4).

**Table 4.** Summary statistics of the items used in evaluating the characteristics of the agricultural sector determinants.

| Female | | | | | |
|---|---|---|---|---|---|
| **Variables** | **Mean** | **Std. Dev.** | **Min** | **Max** | **Coef. of var.** |
| F-OF | 21.86 | 10.88 | 5.19 | 44.9 | 49.77 |
| F-UAA | 13.50 | 6.84 | 2.97 | 27.29 | 50.67 |
| SL.AGR.EMPL.FE.ZS | 3.39 | 4.55 | 0.44 | 22.21 | 134.22 |
| F-FAESAPA | 13.46 | 6.51 | 3.07 | 27.18 | 48.37 |
| F-FWL | 21.25 | 10.67 | 4.73 | 46.48 | 50.21 |
| F-FWLL | 15.56 | 11.76 | 2.61 | 53.15 | 75.58 |
| F-SO | 0.15 | 0.10 | 0.00 | 0.39 | 66.67 |
| F-FWHC > 50% | 18.93 | 18.57 | 0.00 | 50.04 | 98.10 |
| Male | | | | | |
| **Variables** | **Mean** | **Std. Dev.** | **Min** | **Max** | **Coef. of var.** |
| M-OF | 78.14 | 10.88 | 55.1 | 94.81 | 13.92 |
| M-UAA | 86.5 | 6.84 | 72.71 | 97.03 | 7.91 |
| SL.AGR.EMPL.MA.ZS | 5.79 | 4.57 | 1.30 | 22.38 | 78.93 |
| M-FAESAPA | 86.54 | 6.51 | 72.82 | 96.93 | 7.52 |
| M-FWL | 78.75 | 10.67 | 53.52 | 95.27 | 13.55 |
| M-FWLL | 87.29 | 7.78 | 65.29 | 97.46 | 8.91 |
| M-SO | 83.84 | 18.07 | 0.00 | 97.62 | 21.55 |
| M-FWHC > 50% | 40.33 | 35.36 | 0.00 | 95.59 | 87.68 |

Source: own calculations based on Eurostat and Word Bank data.

*4.2. PCA and MRMs Results*

PCA and MRM were applied in order to bring out the differences between male- and female-operated farms in the 27 EU countries, and in order to respond to the RQ$_3$ (*To what extent are the characteristics of farms run by women affected by different socio-economic and labour market determinants?*). The analysis of the main PCs highlighted the differences in the variables of the gender gap in the 27 countries of the EU, and led to the identification, based on the Kaiser criterion theory, of three main components for women and four components for men. Overall, the three components accounted for more than 70% of the total variability for women and the four components accounted for 77% of that for men. There was a low information loss of 30% and 23%, respectively (Table 5).

**Table 5.** PCA and the total of variance explained (female vs. male).

| PC | Eigenvalue | Difference | Proportion | Cumulative |
|---|---|---|---|---|
| Female | | | | |
| PC$_1$ | 5.558 | 2.258 | 0.347 | 0.347 |
| PC$_2$ | 3.299 | 0.533 | 0.206 | 0.554 |
| PC$_3$ | 2.767 | 1.784 | 0.173 | 0.727 |
| Male | | | | |
| PC$_1$ | 4.741 | 1.291 | 0.296 | 0.296 |
| PC$_2$ | 3.449 | 0.465 | 0.216 | 0.512 |
| PC$_3$ | 2.985 | 1.777 | 0.187 | 0.699 |
| PC$_4$ | 1.208 | 0.296 | 0.076 | 0.774 |

Source: own calculations based on Eurostat and Word Bank data.

Table 6 contains the loadings of every variable of the retained components (score coefficients). To interpret the meaning of every factor, the variables that had the greatest loadings on one factor were analysed in terms of their similarity regarding the measured construct. Following this, it was possible to label the PCs according to their relevant meaning. Significant loadings on a PC are defined as those with a loading greater than 0.30 in absolute value. The higher the loading of a variable, the more contribution is reflected by

that variable within a particular PC. The PCA suggested three components with positive signs for the situation concerning women and four for that of men, which meant that the three and the four latent dimensions in the component space accounted for 73% and 77% of the variance, respectively. The first component identified (the presence of educated women in industrial and market-oriented agriculture vs. the presence of rural males in industrial and market-oriented agriculture) included three items concerning the situation for women (SL.IND.EMPL.FE.ZS, F-EA.TE and NV.AGR.TOTL.ZS) and three for the situation concerning men (SL.IND.EMPL.MA.ZS, SP.RUR.TOTL.ZS and NV.AGR.TOTL.ZS). The second component emphasised the presence of women with wages or salaries in the agricultural sector vs. the presence of educated men in permanent cropland, and included one item (SL.EMP.WORK.FE.ZS) for the former and four items for the latter (AG.LND.CROP.ZS, SL.UEM.TOTL.MA.ZS, M-EA.TE and MLPR). The third component, the female use of agricultural land vs. the male use of agricultural and arable land under cereal production, included one item for the situation concerning women and related to the share of agricultural land on total land (AG.LND.AGRI.ZS) and three items concerning the situation for men, relating to the share of agricultural land on the total land in the country (AG.LND.AGRI.ZS), the share of arable land on total land (AG.LND.ARBL.ZS), and the share of land under cereal production on total land (AG.LND.CREL.ZS). The fourth component, the presence of male unemployment in the market-oriented agriculture sector, included two items: SL.UEM.TOTL.MA.ZS and NV.AGR.TOTL.ZS. However, in the countries where more women are employed in the industrial sector, the percentage of value added derived from the agricultural sector is higher. This suggests, in line with the European Parliament (2019), that well-educated women are deciding to move to the countryside to carry out their professional activities.

**Table 6.** Scoring coefficient derived from rotated factor matrix characterising the first main components (f vs. m).

| | Female | | | | Male | | | |
|---|---|---|---|---|---|---|---|---|
| **Variable** | **PC₁** | **PC₂** | **PC₃** | **Variable** | **PC₁** | **PC₂** | **PC₃** | **PC₄** |
| GEI | −0.326 | 0.144 | −0.011 | GEI | −0.315 | −0.233 | −0.070 | 0.009 |
| F-EA.TE | **0.327 *** | 0.181 | −0.123 | M-EA.TE | −0.244 | **0.308** | 0.092 | 0.164 |
| FLPR | −0.142 | 0.238 | −0.265 | MLPR | 0.053 | **0.306** | 0.220 | −0.466 |
| SL.FAM.WORK.FE.ZS | 0.265 | −0.218 | 0.090 | SL.FAM.WORK.MA.ZS | 0.205 | 0.279 | 0.015 | −0.176 |
| SL.IND.EMPL.FE.ZS | **0.350 *** | 0.099 | −0.178 | SL.IND.EMPL.MA.ZS | **0.357 *** | −0.123 | −0.117 | −0.169 |
| SL.SRV.EMPL.FE.ZS | −0.398 | 0.054 | 0.111 | SL.SRV.EMPL.MA.ZS | −0.429 | −0.026 | 0.100 | 0.071 |
| SL.EMP.WORK.FE.ZS | −0.197 | **0.417 *** | −0.142 | SL.EMP.WORK.MA.ZS | −0.052 | −0.457 | −0.029 | 0.255 |
| SL.UEM.TOTL.FE.ZS | −0.054 | −0.455 | 0.063 | SL.UEM.TOTL.MA.ZS | −0.115 | **0.357** | −0.143 | **0.469 *** |
| SP.DYN.LE00.FE.IN | −0.321 | −0.230 | 0.072 | SP.DYN.LE00.MA.IN | −0.392 | 0.083 | 0.010 | −0.282 |
| SP.RUR.TOTL.ZS | 0.270 | −0.059 | −0.267 | SP.RUR.TOTL.ZS | **0.337 *** | 0.095 | −0.179 | −0.162 |
| AG.LND.AGRI.ZS | 0.108 | 0.098 | **0.467 *** | AG.LND.AGRI.ZS | 0.041 | 0.027 | **0.479 *** | 0.053 |
| AG.LND.ARBL.ZS | 0.186 | 0.260 | 0.397 | AG.LND.ARBL.ZS | 0.156 | −0.121 | **0.480 *** | 0.225 |
| AG.LND.CREL.ZS | 0.239 | 0.264 | 0.293 | AG.LND.CREL.ZS | 0.242 | −0.129 | **0.400 *** | 0.219 |
| AG.LND.CROP.ZS | −0.010 | −0.454 | 0.144 | AG.LND.CROP.ZS | −0.092 | **0.461** | 0.014 | 0.191 |
| AG.LND.FRST.K3 | −0.006 | −0.038 | −0.510 | AG.LND.FRST.K3 | 0.090 | −0.081 | −0.481 | 0.157 |
| NV.AGR.TOTL.ZS | **0.309 *** | −0.198 | −0.139 | NV.AGR.TOTL.ZS | **0.317 *** | 0.245 | −0.096 | ***** |

\* PC's loading > 0.30. Source: own calculations based on Eurostat and Word Bank data.

In Table 7, the new latent factors and their denomination are reported.

As seen in Tables 8 and 9, PC₁ is the most important independent variable that characterises female and male roles in the agricultural sector in the best way. In contrast, the *t*-Test was used to assess the significance of the regression coefficients and showed that PC₁ is statistically significant for almost all the dependent variables relating to the characteristics of the agricultural sector ($p < 0.05$). According to the t-statistic values, the item that most affects the role of women and men in the agricultural sector is their presence in terms of employment in this sector. The principal difference between women and men is that for the former the most important percentage is that relating to farms owned, while for the latter, it is that relating to the management of farms with livestock. Indeed, for women,

all variables that describe the characteristics of the agricultural sector in the 27 EU countries depend positively on the Presence of educated females in industrial and market-oriented agriculture (Table 8). In contrast, regarding the situation for men, only the variables SL.AGR.EMPL.MA.ZS and M-FWHC > 50% depend positively on the PC$_1$ (presence of rural male in the industrial and in the market-oriented agriculture). Regarding the second PC, for woman, the relationship between the presence of women with wages and salaries in the agricultural sector and the dependent variables is significant to a small degree and only in relation to F-FWLL. The independent variable SL.AGR.EMPL.MA.ZS is, however, statistically significant for the presence of educated men in the permanent cropland. The female use of agricultural land (PC$_3$) is influenced to a small degree only by the share of women employed in the agricultural sector, while almost all the dependent variables (except SL.AGR.EMPL.MA.ZS and M-FWHC > 50%) are positively correlated with the male use of agricultural and arable land under cereal production, especially the M-FWLL. Finally, concerning the situation for men, according to t-statistic values, the presence of male employment in the market-oriented agricultural sector is positively affected by only two dependent variables: SL.AGR.EMPL.MA.ZS and M-SO (Table 9).

**Table 7.** Denomination of the new latent factors with the percentage of variance (female vs. male).

| Female | | |
|---|---|---|
| **PC** | **Denomination** | **Variance (%)** |
| PC$_1$ | Presence of educated females in industrial and market-oriented agriculture | 35 |
| PC$_2$ | Presence of female waged and salaried workers in the agricultural sector | 21 |
| PC$_3$ | The female use of agricultural land | 17 |
| Male | | |
| PC$_1$ | Presence of rural males in industrial and market-oriented agriculture | 30 |
| PC$_2$ | Presence of educated males in permanent cropland | 22 |
| PC$_3$ | The male use of agricultural and arable land under cereal production | 19 |
| PC$_4$ | The presence of male unemployment in market-oriented agriculture | 8 |

Source: own calculations based on Eurostat and Word Bank data.

**Table 8.** Estimate of coefficients of the multivariate regression (female).

| Equation | RMSE | R-sq | F | *p* |
|---|---|---|---|---|
| **F-OF** | **8.92** | **0.40** | **5.22** | **0.00 \*** |
| F-UAA | 5.73 | 0.38 | 4.69 | 0.01 |
| **SL.AGR.EMPL.FE.ZS** | **2.65** | **0.70** | **1.79** | **0.00 \*** |
| F-FAESAPA | 5.45 | 0.38 | 4.71 | 0.01 |
| F-FWL | 8.87 | 0.39 | 4.88 | 0.00 |
| F-FWLL | 10.50 | 0.29 | 3.21 | 0.04 |
| F-SO | 0.09 | 0.33 | 3.67 | 0.03 |
| F-FWHC > 50% | 15.43 | 0.39 | 4.88 | 0.00 |
| | **Coef** | **Std. Err.** | **t** | ***p* > ItI** |
| **F-OF** | | | | |
| PC$_1$ | 2.22 | 0.74 | 2.99 | 0.01 |
| PC$_2$ | −1.33 | 0.96 | −1.38 | 0.18 |
| PC$_3$ | −2.31 | 1.05 | −2.19 | 0.04 |
| _cons | 2.19 | 1.72 | 12.73 | 0.00 |
| **F-UAA** | | | | |
| PC$_1$ | 1.34 | 0.48 | 2.82 | 0.01 |
| PC$_2$ | −1.11 | 0.62 | −1.79 | 0.09 |
| PC$_3$ | −1.15 | 0.68 | −1.71 | 0.10 |
| _cons | 13.50 | 1.10 | 12.24 | 0.00 |

**Table 8.** *Cont.*

| Equation | RMSE | R-sq | F | *p* |
|---|---|---|---|---|
| **SL.AGR.EMPL.FE.ZS** | | | | |
| PC$_1$ | 1.40 | 0.22 | 6.35 | 0.00 |
| PC$_2$ | −1.04 | 0.29 | −3.64 | 0.00 |
| PC$_3$ | 0.08 | 0.31 | 0.26 | 0.79 |
| _cons | 3.39 | 0.51 | 6.64 | 0.00 |
| **F-FAESAPA** | | | | |
| PC$_1$ | 1.13 | 0.45 | 2.49 | 0.02 |
| PC$_2$ | −1.22 | 0.59 | −2.08 | 0.05 |
| PC$_3$ | −1.23 | 0.64 | −1.91 | 0.07 |
| _cons | 13.46 | 1.05 | 12.84 | 0.00 |
| **F-FWL** | | | | |
| PC$_1$ | 1.88 | 0.74 | 2.55 | 0.02 |
| PC$_2$ | −0.89 | 0.96 | −0.93 | 0.36 |
| PC$_3$ | −2.82 | 1.05 | −2.69 | 0.01 |
| _cons | 21.25 | 1.71 | 12.44 | 0.00 |
| **F-FWLL** | | | | |
| PC$_1$ | 1.63 | 0.87 | 1.86 | 0.08 |
| PC$_2$ | 0.24 | 1.13 | 0.21 | 0.83 |
| PC$_3$ | −3.06 | 1.24 | −2.47 | 0.02 |
| _cons | 15.56 | 2.02 | 7.70 | 0.00 |
| **F-SO** | | | | |
| PC$_1$ | 0.02 | 0.00 | 2.32 | 0.03 |
| PC$_2$ | −0.01 | 0.00 | −1.07 | 0.29 |
| PC$_3$ | −0.02 | 0.01 | −2.11 | 0.05 |
| _cons | 0.15 | 0.02 | 8.98 | 0.00 |
| **F-FWHC > 50%** | | | | |
| PC$_1$ | 3.49 | 1.28 | 2.72 | 0.01 |
| PC$_2$ | −4.08 | 1.67 | −2.45 | 0.02 |
| PC$_3$ | −2.05 | 1.82 | −1.13 | 0.27 |
| _cons | 18.93 | 2.97 | 6.37 | 0.00 |

* $p < 0.05$. Source: own calculations based on Eurostat and Word Bank data.

**Table 9.** Estimate of coefficients of the multivariate regression (male).

| Equation | RMSE | R-sq | F | *p* |
|---|---|---|---|---|
| M-OF | 7.16 | 0.63 | 9.49 | 0.00 |
| M-UAA | 5.06 | 0.54 | 6.40 | 0.00 |
| **SL.AGR.EMPL.MA.ZS** | **2.82** | **0.68** | **1.16** | **0.00 *** |
| M-FAESAPA | 4.72 | 0.55 | 6.85 | 0.00 |
| **M-FWL** | **6.62** | **0.68** | **1.14** | **0.00 *** |
| M-FWLL | 5.59 | 0.56 | 7.09 | 0.00 |
| M-SO | 17.66 | 0.19 | 1.30 | 0.30 |
| M-FWHC > 50% | 29.91 | 0.39 | 3.58 | 0.02 |

| | Coef | Std. Err. | t | *p* > ItI |
|---|---|---|---|---|
| **M-OF** | | | | |
| PC$_1$ | −2.74 | 0.65 | −4.24 | 0.00 |
| PC$_2$ | −1.53 | 0.76 | −2.02 | 0.06 |
| PC$_3$ | 1.85 | 0.81 | 2.27 | 0.03 |
| PC$_4$ | −4.19 | 1.28 | −3.28 | 0.00 |
| _cons | 78.14 | 1.38 | 56.68 | 0.00 |

**Table 9.** *Cont.*

| Equation | RMSE | R-sq | F | *p* |
|---|---|---|---|---|
| **M-UAA** | | | | |
| $PC_1$ | −1.55 | 0.46 | −3.40 | 0.00 |
| $PC_2$ | −1.24 | 0.53 | −2.32 | 0.03 |
| $PC_3$ | 0.99 | 0.57 | 1.72 | 0.10 |
| $PC_4$ | −2.15 | 0.90 | −2.39 | 0.03 |
| _cons | 86.50 | 0.97 | 88.83 | 0.00 |
| **SL.AGR.EMPL.MA.ZS** | | | | |
| $PC_1$ | 1.33 | 0.25 | 5.24 | 0.00 |
| $PC_2$ | 1.24 | 0.30 | 4.17 | 0.00 |
| $PC_3$ | −0.03 | 0.32 | −0.11 | 0.92 |
| $PC_4$ | 0.67 | 0.50 | 1.32 | 0.20 |
| _cons | 5.79 | 0.54 | 10.68 | 0.00 |
| **M-FAESAPA** | | | | |
| $PC_1$ | −1.32 | 0.43 | −3.09 | 0.00 |
| $PC_2$ | −1.25 | 0.50 | −2.51 | 0.02 |
| $PC_3$ | 1.15 | 0.54 | 2.14 | 0.04 |
| $PC_4$ | −2.22 | 0.84 | −2.64 | 0.02 |
| _cons | 86.54 | 0.91 | 95.23 | 0.00 |
| **M-FWL** | | | | |
| $PC_1$ | −2.57 | 0.60 | −4.31 | 0.00 |
| $PC_2$ | −0.89 | 0.70 | −1.28 | 0.22 |
| $PC_3$ | 2.29 | 0.75 | 3.06 | 0.01 |
| $PC_4$ | −4.74 | 1.18 | −4.02 | 0.00 |
| _cons | 78.75 | 1.27 | 61.85 | 0.00 |
| **M-FWLL** | | | | |
| $PC_1$ | −1.70 | 0.50 | −3.38 | 0.00 |
| $PC_2$ | 0.01 | 0.59 | 0.02 | 0.99 |
| $PC_3$ | 1.58 | 0.63 | 2.49 | 0.02 |
| $PC_4$ | −3.27 | 0.99 | −3.28 | 0.00 |
| _cons | 87.29 | 1.08 | 81.16 | 0.00 |
| **M-SO** | | | | |
| $PC_1$ | −3.46 | 1.59 | −2.17 | 0.04 |
| $PC_2$ | 0.24 | 1.86 | 0.13 | 0.90 |
| $PC_3$ | 0.65 | 2.00 | 0.33 | 0.75 |
| $PC_4$ | 1.88 | 3.15 | 0.60 | 0.56 |
| _cons | 83.84 | 3.40 | 24.67 | 0.00 |
| **M-FWHC > 50%** | | | | |
| $PC_1$ | 6.71 | 2.69 | 2.49 | 0.02 |
| $PC_2$ | 8.94 | 3.16 | 2.83 | 0.01 |
| $PC_3$ | −0.53 | 3.40 | −0.16 | 0.88 |
| $PC_4$ | −1.67 | 5.34 | −0.31 | 0.76 |
| _cons | 40.33 | 5.76 | 7.01 | 0.00 |

* $p < 0.05$. Source: own calculations based on Eurostat and Word Bank data.

## 5. Discussion, Conclusions, and Practical Implications

This study sought to provide an overall picture of the key determinants that affect the differences between farms operated by women and those operated by men in the agricultural sector. The study was underpinned by four research questions, which assumed that the interaction effects of socio-economic and labour market determinants ($RQ_1$) would be likely to provide evidence of differences between women and men in agricultural land use ($RQ_2$); would influence the results of female-run farms when compared with male-run farms ($RQ_3$); and would require reforming laws to guarantee equal rights, encourage female empowerment, and ensure that women are aware of their rights and able to claim them ($RQ_4$). In order to validate the above RQs, an empirical analysis was therefore performed

on the open data drawn from the World Bank and Eurostat sites by combining DSA, PCA, and MRMs. In detail, the findings of this analysis, in line with the World Bank (2012), show that differences between female- and male-operated farms are still influenced by some important determinants.

In descriptive terms, the number of farms operated by both women and men decreased between 2005 and 2016 in almost all EU countries, with the exception of Luxembourg and Sweden for the former and Ireland for the latter. However, male-run farms (−31%) contributed more to this decline than those run by females (−23%). Approximately 30% of farms across the EU-27 are managed by women; countries with the highest share of female farm managers are Latvia and Lithuania, while in other member states the proportion is below the EU average (Germany, Denmark, Malta, and The Netherlands).

How this will impact the gender gap in the agricultural sector is open to debate. Academic researchers in the agricultural sector are only starting to study the various determinants that have affected this decline and these differences. With the aim of filling this research gap, for the present study, the most influential determinates were drawn from the few studies on the subject, the World Bank Open Data repository, and the Eurostat database. In this regard, the findings of the analysis provide the following answers to the first three RQs:

Firstly, the DSA reveals that, on average, women are more educated than their male counterparts, but their participation in the labour market is lower. They also have fewer employment opportunities compared with men in rural areas. However, data show that female workers, on average, are more often salaried than their male counterparts. Regarding the agricultural sector, the results reveal that, on average, more than three quarters of farms are run by men. These are larger farms than those run by women as they incorporate approximately 86% of the agricultural area currently used for production purposes. Differences in the average size of the farmland owned by women and men are also reflected in the output per holding: women farmers fare much less well than their male counterparts

Secondly, on average, at the EU level, the employment position of women is better in the Northern and Western member states compared with the Southern and Central-Eastern member states. This has often been explained by a more conservative gender contract in the Mediterranean, as well as Eastern European countries, reflected in less political support for family and work reconciliation services. The economic strength of member states also plays an important role in the availability of employment in the tertiary sector of the labour market. Each RQ under study has revealed differences between men and women, which are linked to the various explanatory variables adopted in Models 1 and 2.

Indeed, the results show that the socio-economic determinates and the labour market variables that most affect the characteristics of agricultural farms are the presence of women and men in terms employment; the percentage of value added derived from the agricultural sector; and the level of education. This is consistent with the findings of previous studies, which show that the productive value of education, and the professional activities that women in particular have accrued by working in the industrial sector, have two main effects on the value added to the agricultural sector: the "worker effect" and the "transfer effect" (European Parliament 2019). Worker effects are described as the situation whereby an educated farmer, given the same input, can produce a greater output that makes better use of current resources (Welch 1970). With the "transfer effect", educated women are deciding to move to the countryside to carry out their professional activities. This suggests that education is a crucial factor for bridging the gender gap in the agricultural sector—in particular gender equality in education is crucial for fostering agricultural growth. The values of the coefficient of variation highlight the existence of more heterogeneity of female education across EU countries in respect to levels of male education. There is also a correlation between the gender equality index and female life expectancy at birth. This confirms that equality between women and men is vital for the economic and social growth of a country. From the point of view of practical implications, Table 10 reports some

suggestions to respond to the fourth and final RQ$_4$: how to bridge the gender gap in the EU agricultural sector?

**Table 10.** How to bridge the gender gap in the EU agricultural sector in relation to assets of determinants considered.

| Assets of Determinants Considered | Female vs. Male | Some Suggestions to Bridge the Gender Gap |
| --- | --- | --- |
| High education | On average, females exceed male attainment levels (44% vs. 25%), but in some countries women still lag behind (Italy, Malta, Portugal, and Spain). | Education is an important drive to increase women's chances to enter the industrial sector and then to move into the agricultural sector to transfer their acquired knowledge and to access rural wage employment. However, the emphasis of education policies should evidently vary depending on whether the country's labour market is dominated by agricultural activities or non-agricultural activities. |
| Labour markets | On average, the diverse share of male vs. female in the sectors of the economy highlight the women who tend to be concentrated in the tertiary sector (84% vs. 61%). | This is due probably to the improvements in EU policies for public and private sector organisations. The increasing level of educational activities among women has presented a strong association between high education and participation of females in the industrial sector. |
| Care family workers | On average, women have heavy and unpaid household duties that take them away from more productive activities when compared with men (1.32% vs. 0.67%). | Women in EU countries tend to earn less than men as they do not get time to set up activities in the market-oriented agricultural sector and, often, they work in this sector as waged or salaried workers. |
| Agricultural labour market | On average agriculture is much more important for men than for women in terms of the percentage of employment (5.79% vs. 3.39%). | Provide women with proper guidance and assistance to deal with their own farm set ups or to start their own organisations. Women should be treated as individual farmers, not just as part of the household labour force. |
| Land use and standard output | On average, more than three quarters of farms are run by men (78%). These are larger farms than those run by women as they incorporate on average more than 86% of the agricultural area currently used for production purposes (UAA). Differences between men and women in the average size of farmland owned are also reflected in the output per holding: women farmers also fare much less well than their male counterparts (84% vs. 0.15%). | Closing the gap in access to land and other agricultural assets requires, among other things, reforming laws to guarantee equal rights, educating government officials and community leaders and holding them accountable for upholding the law. It also involves empowering women to ensure that they are aware of their rights and are able to claim them. |

Source: own calculations based on Eurostat and Word Bank data.

An outline of the general picture of the most influential determinants that affect the gender gap in the agricultural sector can both inspire future inquiries and assist the EU by promoting greater political support for strengthening the role of women in the agricultural sector. This is in consideration of the fact that agriculture can be an important engine of growth and poverty reduction. However, the sector is underperforming in many countries in part because women, who are often a crucial resource in agriculture and the rural economy, face constraints that reduce their productivity. Furthermore, the analysis may be considered as a further step towards a more visible role for women in agriculture, as opposed to the subsidiary/invisible role underlined in past decades.

Author should discuss the results and how they can be interpreted from the perspective of previous studies and of the working hypotheses. The findings and their implications

should be discussed in the broadest context possible. Future research directions may also be highlighted.

**Funding:** This research received no external funding.

**Institutional Review Board Statement:** Not applicable.

**Data Availability Statement:** World Bank (2019). Available at: https://data.worldbank.org/indicator FAO. Accessed on 20 September 2021.

**Conflicts of Interest:** The author declares no conflict of interest.

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
