# Peer review of "Bridging the Gender Gap in the Agricultural Sector: Evidence from European Union Countries"

_socsci, doi:10.3390/socsci11030105_

Round 1

Reviewer 1 Report

The study presented statistical analysis regarding the differences between female and male-operated farms in EU countries. The manuscript, in my opinion, is informative, insightful, and very well-written. The authors did a great job on presenting their results and discussing the statistics in depth. I only have a few trivial observations and do not have any major comments. I recommend the manuscript in its current form for publication.

  • Line 269, period missing.
  • Table 1, maybe spell out “var %”.
  • Figure 1, Latvia and Lithuania stand out from the rest of the countries by a substantial margin. If possible, I encourage the authors to investigate further for potential reasons or present hypotheses that might explain the phenomenon.
  • Line 543-546 editorial mistake.

The reason why I selected "minor" was because the manuscript has a few editing errors that need to be corrected. The manuscript itself was very well written in my opinion. After reading the manuscript, I simply did not observe anything major that deserves my comments, since the authors told a very complete story. Please do not think that I did not do a careful review just because my comments were short. The authors are far more experts than me on the specific topic.

Author Response

Responses to reviewer 1

Thank you for the valuable observations and useful suggestions; I have tried to answer to all the issues raised at the best of my knowledge, and below our point-by-point responses are reported.

The study presented statistical analysis regarding the differences between female and male-operated farms in EU countries. The manuscript, in my opinion, is informative, insightful, and very well-written. The authors did a great job on presenting their results and discussing the statistics in depth. I only have a few trivial observations and do not have any major comments. I recommend the manuscript in its current form for publication.

  • Line 269, period missing.

In the revised version of the paper please see red lines 288-290: Male-run farms (-31%) rather than female-run farms (-23%), contributed more to this de-cline. It is possible to hypothesize that in the period 2005-2016 farms run by women have shown greater resilience compared to those run by men.

  • Table 1, maybe spell out “var %”.

Lines 292-293: Table 1 - Female vs male operated farms in the EU countries, var % 2016-2005

  • Figure 1, Latvia and Lithuania stand out from the rest of the countries by a substantial margin. If possible, I encourage the authors to investigate further for potential reasons or present hypotheses that might explain the phenomenon.

See red lines 296-309: As can be seen in Figure 1, the EU countries with a higher incidence of female-run farms (>30%) on the total number of farms are in order of importance: Latvia, Lithuania, Romania, Estonia, Austria, Italy and Portugal. Conversely, those with a low incidence (<10%) are the Netherlands, Malta, Denmark and Germany. This can be attributed to the fact that in the first block of countries, especially in Latvia and Lithuania, subsistence agriculture prevails and farms encounter the marginalisation of productive resources and human capital; while in the second block, agriculture with market-oriented farms prevails, has higher incomes and is led mainly by men. Furthermore, these trends are related not only to cultural factors but also and, above all, to the type of agricultural activity present in the various geographical areas. So much so that the states with the lowest quotas of female management (<10%) are all characterised by a strong zootechnical specialisation. This has traditionally been the prerogative of the male gender, since the establishment of the first complex and technologically advanced communities ("intensive agriculture passes into the hands of men and breeding is always a male task” (Arioti, 1983).

Line 543-546 editorial mistake.

See red lines 603-613: Author should discuss the results and how they can be interpreted from the perspective of previous studies and of the working hypotheses. The findings and their implications should be discussed in the broadest context possible. Future research directions may also be highlighted.

Funding: This research received no external funding

Data Availability Statement: World Bank (2019). Available at: https://data.worldbank.org/indicator. FAO

Conflicts of Interest: The author declares no conflict of interest

The reason why I selected "minor" was because the manuscript has a few editing errors that need to be corrected. The manuscript itself was very well written in my opinion. After reading the manuscript, I simply did not observe anything major that deserves my comments, since the authors told a very complete story. Please do not think that I did not do a careful review just because my comments were short. The authors are far more experts than me on the specific topic.

Thank you a lot again

Reviewer 2 Report

Significance of the Study: 

Why this study is significant in countries where 

a) Majority of the women are educated and in service sectors

b) In countries that import foods 

c) In countries where corporations produce foods as a cash crop and fulfill the needs of other countries

The authr/s need to provide information on 

A. what percentages of women live in the rural area of the countries they have studied

 B. What percentages of them are in agricultural work? 

C. Why they need to emerge as agricultural entrepreneurs.

This information is needed to establish the importance of the study.

Theory

The researchers may consider using the Time reassignment theory and Shelby Walby's Patriarchal theories to explain Women's employment in agriculture and work as family workers.

Methodology:

Variable names are unclear, and a reader has to go back and forth to remember the variable names. For example, instead of SLAGR, can it be f/m employee Agri.

Findings:

Data Presentation is very scattered. I had to go back and forward to remember the name of the variables and relate them with the coefficient value, significant level, and r square value. At present, tables have too much info. Authors can provide only coefficient value, significant level, F value, and r square value in the main table and have other information in the appendix. P-value can be expressed using the * sign beside the coefficient values. That will reduce one column. ( Table 8 and 9)

Authors need to explain why the SLAGR model has the most explaining power ( based on adjusted R square value) than the other model. Authors may use other literature to explain that. Also, think of policy implications based on the strength of the model. 

Instead of having RQ in the findings, have a heading. Then discuss the RQ and the results. 

Author Response

Responses to reviewer 2

Thank you for the valuable observations and useful suggestions; I have tried to answer to all the issues raised at the best of my knowledge, and below our point-by-point responses are reported.

Significance of the Study: 

Please see in revised version of the manuscript the first red lines of the introduction:

The principal aim of this paper is to present the results of a study into the gender gap (gender inequalities) that influence the characteristics of the agricultural sector. It focusses specifically on how the sector is affected by the different determinants of the economic systems of European Union (EU) countries and by the diverse agricultural land use, which very little is known about.

Why this study is significant in countries where 

  1. a) Majority of the women are educated and in service sectors
  2. b) In countries that import foods 
  3. c) In countries where corporations produce foods as a cash crop and fulfill the needs of other countries
  4. a) Please see in revised version of the manuscript red lines 137-139: In recent years, in some EU regions a new phenomenon has come about, which shows that well-educated women are deciding to move to the countryside to carry out their professional activities (European Parliament, 2019).

b+c) Please see in revised version of the manuscript red lines 296-310: As can be seen in Figure 1, the EU countries with a higher incidence of female-run farms (>30%) on the total number of farms are in order of importance: Latvia, Lithuania, Romania, Estonia, Austria, Italy and Portugal. Conversely, those with a low incidence (<10%) are the Netherlands, Malta, Denmark and Germany. This can be attributed to the fact that in the first block of countries, especially in Latvia and Lithuania, subsistence agriculture prevails and farms encounter the marginalisation of productive resources and human capital; while in the second block, agriculture with market-oriented farms prevails, has higher incomes and is led mainly by men. Furthermore, these trends are related not only to cultural factors but also and, above all, to the type of agricultural activity present in the various geographical areas. So much so that the states with the lowest quotas of female management (<10%) are all characterised by a strong zootechnical specialisation. This has traditionally been the prerogative of the male gender, since the establishment of the first complex and technologically advanced communities ("intensive agriculture passes into the hands of men and breeding is always a male task” (Arioti, 1983).

The author/s need to provide information on 

  1. what percentages of women live in the rural area of the countries they have studied

Please see in revised version of the manuscript red lines 45-47: At the EU level (EU-28), around 30% of agricultural farms are operated by women - one farm in five and in rural areas of the EU women represent over the 50% of the rural population.

  1. What percentages of them are in agricultural work? 

Please see in revised version of the manuscript red lines 48-52: Women comprised about 45% of the total working population and about 35% of workers in the agricultural sector of the EU-28, the work of female farmers accounts for 31% of the total working hours (European Parliament, 2019). Despite this importance, little is known about the differences between the characteristics of farms managed by women and those managed by men.

  1. Why they need to emerge as agricultural entrepreneurs.

Please see in revised version of the manuscript red lines 593-606: An outline of the general picture of the most influential determinants that affect the gender gap in the agricultural sector can both inspire future inquiries and assist the EU by promoting greater political support for strengthening the role of women in the agricultural sector. This in consideration of the fact that agriculture can be an important engine of growth and poverty reduction. But the sector is underperforming in many countries in part because women, who are often a crucial resource in agriculture and the rural economy, face constraints that reduce their productivity. Furthermore, the analysis may be considered as a further step towards a more visible role for women in agriculture, as opposed to the subsidiary/invisible role underlined in past decades.

Author should discuss the results and how they can be interpreted from the perspective of previous studies and of the working hypotheses. The findings and their implications should be discussed in the broadest context possible. Future research directions may also be highlighted.

This information is needed to establish the importance of the study.

Theory

The researchers may consider using the Time reassignment theory and Shelby Walby's Patriarchal theories to explain Women's employment in agriculture and work as family workers.

Thank you for your suggestions but in my article I have not considered these theories because my main aim is to consider the main factors that explain why the difference between men and women still persists in the agricultural sector. One of this indicator is the the synthetic index on gender equality (GEI). If there were gender equality in a country this would encourage women to participate in higher wage-earning sectors such as the industrial and services sectors rather than in the agricultural sector. Special attention is paid to the interactions between men and women in relation to male and female participation and unemployment. It might be that men and women compete for the same jobs but it could also be that the labour market status of men (especially with regard to male unemployment) has a discouraging effect on the behaviour of women or vice versa.

Methodology:

Variable names are unclear, and a reader has to go back and forth to remember the variable names. For example, instead of SLAGR, can it be f/m employee Agri.

The extended name of the variables and the acronym of each of them in brackets are shown in the subsections 3.1.1, 3.1.2 and 3.1.3.

Findings:

Data Presentation is very scattered. I had to go back and forward to remember the name of the variables and relate them with the coefficient value, significant level, and r square value. At present, tables have too much info. Authors can provide only coefficient value, significant level, F value, and r square value in the main table and have other information in the appendix. P-value can be expressed using the * sign beside the coefficient values. That will reduce one column. (Table 8 and 9).

Please see the adjusted tables 8 and 9.

Authors need to explain why the SLAGR model has the most explaining power (based on adjusted R square value) than the other model. Authors may use other literature to explain that. Also, think of policy implications based on the strength of the model. 

Please see in revised version of the manuscript red lines 513-586: This study sought to provide an overall picture of the key determinants that affect the differences between farms operated by women and those operated by men in the agricultural sector. The study was underpinned by four research questions, which assumed that the interaction effects of socio-economic and labour market determinants (RQ1) would be likely to provide evidence of differences between women and men in agricultural land use (RQ2); would influence the results of female-run farms when compared to male-run farms (RQ3); and would require reforming laws to guarantee equal rights, encourage female empowerment and ensure that women are aware of their rights and able to claim them (RQ4). In order to validate the above RQs, an empirical analysis was therefore performed on the open data drawn from World Bank and Eurostat sites by combining DSA, PCA and MRMs. In detail, the findings of this analysis, in line with the World Bank (2012), show that differences between female and male operated farms are still influenced by some important determinants.

In descriptive terms, the number of farms operated by both women and men decreased between 2005 to 2016 in almost all EU countries, with the exception of Luxembourg and Sweden for the former and Ireland for the latter. However, male-run farms (-31%) contributed more to this decline than those run by females (-23%). Approximately 30% of farms across the EU-27 are managed by women; countries with the highest share of female farm managers are Latvia and Lithuania, while in other member states the proportion is below the EU average (Germany, Denmark, Malta, The Netherlands).

How this will impact the gender gap in the agricultural sector is open to debate. Academic researchers in the agricultural sector are only starting to study the various determinants that have affected this decline and these differences. With the aim of filling this research gap, for the present study the most influential determinates were drawn from the few studies on the subject, the World Bank Open Data repository and the Eurostat database. In this regard, the findings of the analyzes provided the following answers to the first three RQs:

RQ1: Are the interaction effects of the socio-economic and labour market determinants on the gender gap in the agricultural sector significant?

RQ2: Are there significant differences in the use of agricultural land between EU countries?

RQ3: To what extent are the characteristics of farms run by women affected by different socio-economic and labour market determinants?

Firstly, the DSA reveals that, on average, women are more educated than their male counterparts but their participation in the labour market is lower. They also have fewer employment opportunities compared to men in rural areas. However, data show that female workers, on average, are more often salaried than their male counterparts. Regarding the agricultural sector, the results reveal that, on average, more than three quarters of farms are run by men. These are larger farms than those run by women as they incorporate approximately 86% of the agricultural area currently used for production purposes. Differences in the average size of the farmland owned by women and men are also reflected in the output per holding: women farmers fare much less well than their male counterparts

Secondly, on average, at the EU level, the employment position of women is better in the Northern and Western member states compared to the Southern and Central-Eastern member states. This has often been explained by a more conservative gender contract in the Mediterranean, as well as Eastern European countries, reflected in less political support for family and work reconciliation services. The economic strength of member states also plays an important role in the availability of employment in the tertiary sector of the labour market. Each RQ under study has revealed differences between men and women, which are linked to the various explanatory variables adopted in Models 1 and 2.

Indeed, the results show that the socio-economic determinates and the labour market variables that most affect the characteristics of agricultural farms are the presence of women and men in terms employment; the percentage of value added derived from the agricultural sector; and the level of education. This is consistent with the findings of previous studies, which have shown that the productive value of education and the professional activities that women in particular, have accrued by working in the industrial sector, have two main effects on the value added to the agricultural sector: the “worker effect” and the “transfer effect” (European Parliament, 2019). Worker effects are described as the situation whereby an educated farmer, given the same input, can produce a greater output that makes better use of current resources (Welch, 1970). With the “transfer effect”, educated women are deciding to move to the countryside to carry out their professional activities. This suggests that education is a crucial factor to bridging the gender gap in the agricultural sector - in particular gender equality in education is crucial for fostering agricultural growth. The values of the coefficient of variation highlight the existence of more heterogeneity of female education across EU countries in respect to levels of male education. There is also a correlation between the gender equality index and female life expectancy at birth. This confirms that equality between women and men is vital for the economic and social growth of a country. From the point of view of practical implications, Table 10 reports some suggestions to respond to the fourth and final RQ4: How to bridge the gender gap in the EU agricultural sector? (Please see table 10, lines 588-591).

Thank you very much again for your comments

Kind regards

Round 2

Reviewer 2 Report

The manuscript looks better after the revision. I still believe that the manuscript would have explained the f/m inequality in agriculture if the author/s used Shelby Walby's Patriarchy Theory. However, I will not hold the paper for that.